# Role and Effect of Meso-Structuring Surfactants on Properties and Formation Mechanism of Microfluidic-Enabled Mesoporous Silica Microspheres

**DOI:** 10.3390/mi14050936

**Published:** 2023-04-26

**Authors:** Nizar Bchellaoui, Qisheng Xu, Xuming Zhang, El-Eulmi Bendeif, Rachid Bennacer, Abdel I. El Abed

**Affiliations:** 1Laboratoire Lumière Matière et Interfaces (LUMIN), UMR 9024, Ecole Normale Supérieure Paris Saclay, CentraleSupélec, CNRS, Université Paris-Saclay, 4 Avenue des Sciences, 91190 Gif-sur-Yvette, France; 2Photonics Research Institute, The Hong Kong Polytechnic University, Kowloon, Hong Kong 999077, China; 3CRM2 (UMR UL-CNRS 7036), Faculté des Sciences et Technologies, Université de Lorraine, BP 70239, Boulevard des Aiguillettes, 54506 Vandoeuvre-lès-Nancy CEDEX, 54000 Nancy, France; 4LPMS, ENS Paris Saclay, CentraleSupélec, Université Paris Saclay, CNRS, 4 Avenue des Sciences, 91190 Gif-sur-Yvette, France

**Keywords:** droplets, microfluidics, meso-structuring agent, surfactants, mesopores, microcapsules, silica, condensation, evaporation

## Abstract

We have shown in a previous work that the combination of the emulsion solvent evaporation technique and droplet-based microfluidics allows for the synthesis of well-defined monodisperse mesoporous silica microcapsules (hollow microspheres), whose size, shape and composition may be finely and easily controlled. In this study, we focus on the crucial role played by the popular Pluronic^®^ P123 surfactant, used for controlling the mesoporosity of synthesised silica microparticles. We show in particular, that although both types of initial precursor droplets, prepared with and without P123 meso-structuring agent, namely P123^+^ and P123^−^ droplets, have a similar diameter (≃30 μm) and a similar TEOS silica precursor concentration (0.34 M), the resulting microparticles exhibit two noticeably different sizes and mass densities. Namely, 10 μm and 0.55 g/cm^3^ for P123^+^ microparticles, and 5.2 μm and 1.4 g/cm^3^ for P123^−^ microparticles. To explain such differences, we used optical and scanning electron microscopies, small-angle X-ray diffraction and BET measurements to analyse structural properties of both types of microparticles and show that in the absence of Pluronic molecules, P123^−^ microdroplets divide during their condensation process, on average, into three smaller droplets before condensing into silica solid microspheres with a smaller size and a higher mass density than those obtained in the presence of P123 surfactant molecules. Based on these results and on condensation kinetics analysis, we also propose an original mechanism for the formation of silica microspheres in the presence and in the absence of the meso-structuring and pore-forming P123 molecules.

## 1. Introduction

The synthesis of mesoporous silica microspheres with a controllable nanopore size and structure is of a significant importance in many academic and industrial research fields, such as drug and gene-delivery, catalysis, biosensing and/or tissue bio-engineering [1,2,3,4,5,6,7]. Many synthetic methods have been developed in the past for their synthesis and the control of their porosity. More than a decade ago, Andersson et al. [8,9,10] developed an original method, called the “emulsion solvent evaporation” method (or ESE), which proved to be very effective for the elaboration of well-ordered meso-structured silica microspheres. This method is based on the combination of the well-known Stöber sol-gel technique and the dispersing, in a continuous oil phase, of sol droplets made of a water–ethanol mixture, a silica precursor (TEOS) and a templating surfactant (PEG derivative) [11] and the dispersing in a continuous oil phase of a silica precursor solution (sol) droplets, which consists generally of a tetraethyl orthosilicate (TEOS) or a tetra methoxy orthosilicate (TMOS), a mixture of water/alcohol solvents and a meso-structuring surfactant (generally a PEG derivative or CTAB). The progressive diffusion of alcohol from droplets to the continuous oil phase enables the concentration of the mesostructuring (P123) surfactant to increase inside droplets. This leads in turn to the self-assembling of the meso-structuring surfactant molecules inside droplets and the formation of spherical or cylindrical micellar structures around which silica solidifies [12]. Furthermore, since water does not readily diffuse into the oil phase, this method usually requires further evaporation of water at elevated temperature and/or reduced pressure.

The ESE method suffers, however, from a rather broad size distribution of the synthesised microspheres, owing to the poor control of the distribution size of the templating droplets, which is simply obtained by vigorous mechanical stirring of the emulsion. We have developed, in a previous work [13], a two-step droplet microfluidic approach, which enables the fabrication of well-defined monodisperse mesoporous silica hollow microspheres (microcapsules), whose size, shape and composition can be readily controlled in a very accurate manner. Droplet-based microfluidics have proven to be a straightforward and robust approach to address the size distribution of ESE-based mesoporous microspheres at the microscale [14,15,16] and highly functional monodisperse microcapsules (see for recent reviews References [17,18,19]).

Caroll et al. [14] were the first to achieve fabrication of highly monodisperse silica microspheres from monodisperse sol droplets generated in a microfluidic device and condensed quickly in a flask outside the microfluidic device. Lee et al. [15] developed a one-step microfluidic approach, which enables for a rapid in situ diffusion of the sol solvents in the carrier oil (hexadecane) and a rapid condensation of silica microspheres in the microfluidic device. Chokkalingam et al. [16] developed another one-step approach by triggering a quick condensation of sol droplets inside the microfluidic channels by means of electrodes embedded inside the microfluidic device. In contrast with Caroll’s [14] and Lee’s [15] works, Chokkalingam et al. used a perfluorinated oil (namely, perfluodecalin), but with a high concentration of stabilising droplets surfactants, i.e., 20 % (*w/w*), in order to stabilise droplets against merging. Such a large concentration of surfactant may explain why, although perfluorinated oils are known to solubilise none of the sol components, they observe a large difference between the amount of silica brought initially in microdroplets (in the form of TMOS) and the final amount of silica remaining in the silica microspheres—∼10^−7^ g and ∼10^−12^ g, respectively.

We have shown in particular, in contrast with previously developed microflluidic methods [14,15,16], that the use of a low concentration of silica precursor in highly monodisperse droplets carried by a fluorocarbon oil and a reduced oil–surfactant concentration allows for the fabrication of highly monodisperse hollow mesoporous microspheres without the use of any additional template [13]. A crucial step of our approach relies on the control of the solvent evaporation process which is carried outside the microfluidic channels. The synthesis process is based on solvent extraction in a silica sol microdroplet that is suspended at the interface between the oil and air. The formation of the mesoporous silica shell is driven only by a control of the balance between the solvent diffusion/evaporation and the formation of a thin mesoporous silica skin at the surface of the microdroplets.

We focus in this study on the role played by the Pluronic^®^ P123 meso-structuring surfactant agent and the presence of nanopores in the microcapsules shells on the condensation process of such microcapsules. Our results enable to better understand the mechanism of formation of such highly monodisperse microcapsules as well as their structural properties while using droplet-based microfluidics.

## 2. Material and Methods

Tetraethyl orthosilicate (TEOS) (99 %; Sigma-Aldrich, Saint Louis, MO 63103-2530, USA) was used as a silica source. The Pluronic P123 meso-structuring triblock copolymer was purchased from Sigma-Aldrich, France. It consists of a central hydrophobic poly(propylene-oxide) (PPO) block sandwiched between two hydrophilic poly(ethylene-oxide) (PEO) blocks with a molar weight of about Mn ≃ 5800 g/mol, owing to its chemical architecture: ${\left(EO\right)}_{20}{\left(PO\right)}_{69}{\left(EO\right)}_{20}$. It is generally used for the synthesis of mesoporous silica materials with a well-ordered hexagonal arrangement of mesopores, such as MCM-41 (Mobil Crystalline Materials-41) [20].

The silica sol precursor was prepared by dissolving, under stirring, at 60 °C, 1 g of P123 in 20 mL of 2 M HNO_3_ and 5 mL of distilled water solution, until the solution became clear. Then, 3.6 g of TEOS was added to the solution under vigorous stirring for 3 h. Then we complete the synthesis by the microfluidic method. TEOS concentration in the solution was 7.2% wt. We also synthesised silica samples without adding P123 surfactants in the silica precursor solution.

The microfluidic device was prepared by standard soft lithography technique in poly(dimethylsiloxane) (PDMS). Ref. [21] A flow-focusing geometry was designed and used to produce monodisperse droplets, shown in Figure 1. The 50 μm wide nozzle enabled the introduction of a thin thread of aqueous phase into the main channel, and the formed droplets were collected in a Petri dish. Moreover, the droplet production process was monitored using a standard CCD camera.

A commercially available HFE 7500 fluorinated oil (3-ethoxy-dodecafluoro-2-trifluoromethyl-hexane, Inventec), having a density of 1.62 g/cm^3^, was used as the carrier oil. This fluorinated oil does not cause PDMS swelling and does not solubilise most nonfluorinated organic molecules, including components of the sol droplets. Droplets were stabilised using a home-prepared copolymer surfactant, produced from a commercially available carboxy-terminated fluorinated polymer, namely, Krytox 157-FSH (Dupont) and a solution of benzyl-trimethylammonium hydroxide (BTA, Sigma-Aldrich, Saint Louis, MO 63103-2530, USA). A commercial surface coating agent (fluorosilane) dried with *N*_2_ was used in order to increase the wettability of the oil phase on the channel walls. Volumetric flow rates were set to $Q_{oil}$ = 300 μL/h and $Q_{aq}$ =30 μL/h for the oil phase and the aqueous phase, respectively, for all experiments (Nemesys, Cetoni GmbH, Korbussen, Germany). This set-up obtains highly monodisperse microdroplets with a diameter of about 100 μm. Images of droplets in the micro-channels were taken using a standard CCD camera (uEye, IDS, Boldon, UK) at 30 frames per second. The droplet size distribution and image processing were realised using ImageJ software.

The SEM analysis, performed using a HITACHI S-3400N microscope(Hitachi, Ltd., Tokyo, Japan) with 20 kV high voltage, shows the surface morphology of the synthesised silica microspheres with different sizes of droplets. Porosity of samples was determined using BET (Brunauer, Emmett, and Teller) 16 measurements and a FlowSorb II 2300 apparatus, with Nitrogen as adsorbent and an analysis bath of 77 K. X-ray diffraction measurements were performed using a PanalyticalX’Pert Pro diffractometer equipped with a Cu tube, a Ge (111) incident beam monochromatic (l = 1.5406 Å) and an X’Celerator detector. Small-angle X-ray scattering (SAXS) measurements were collected using 0.02 rad Soller slits, 1/16 degree fixed divergence and anti-scatter slits. The X’Celerator detector was used as scanning line detector (1D) with 0.518 degree active length. Data collection was carried out in the scattering angle range 0.5–6 degrees with a 0.0167-degree step over 60 min.

### Collection and Condensation of Silica Precursor Droplets

Droplets were collected in a Petri dish and dried at a temperature of 150 °C (under normal pressure). Because of the higher mass density of the fluorocarbon oil ($d_{HFE7500}=1.62$) in relation to the sol phase ($d_{sol}≃1$), collected droplets formed a well-organised floating layer on the surface of the fluorocarbon oil, as illustrated in Figure 1. Moreover, because perfluorocarbon oils are chemically inert, they do not solubilise any of the droplet content unless a high concentration of surfactant is used, which is not the case in the present study; diffusion and dispersion of both solvents and sol may be neglected during the flow of droplets along the microfluidic channel and the tubing. In order to emphasise such a feature, we measured and checked that droplet size remained practically constant during the flow along the microfluidic channel and tubing, from the nozzle production area to the collecting Petri dish, i.e., 31.5 μm.

Evaporation and condensation process starts after droplets spread at the oil/air interface in the Petri dish. Solvent contained in droplets starts to evaporate by diffusing in the air upper phase (but not inside the lower fluorocarbon oil subphase). Consequently, the condensation of silica is mainly governed in our study by a slow evaporation process, at room temperature, of the solvents at the oil–air interface. Droplets were allowed to condense during at least 24 h at room temperature and normal pressure before they were dried at a temperature equal at least to 100 °C (under normal pressure). The samples were then analyzed by optical and scanning electron microscopies, small-angle X-ray scattering (SAXS) and BET, as detailed in the next paragraph.

## 3. Results and Discussion

### 3.1. Structural Characterization of Mesoporous and Non-Mesoporous Silica Microparticles

Figure 2 shows scanning electron microscopy (SEM) images of synthesised solid silica microspheres derived from droplets prepared with and without P123 molecules. They are labelled here P123^+^ and P123^−^ microdroplets, respectively. It is worth noting that, although both types of P123^+^ and P123^−^ initial droplets had a similar diameter, 31.5 μm (±1 μm), the resulting microparticles exhibited two noticeably different sizes. For microparticles derived from P123^+^ microdroplets, we obtained highly monodisperse microspheres with a diameter of about 10 μm (±0.5 μm). These microparticles are referred in this study as the MP (or mesoporous) microparticles or microspheres. For microparticles derived from P123^−^ droplets, we obtained microspheres with a diameter of about 5.2 μm (±0.8 μm), as shown in Figure 2B. We refer to them as the NMP (or non-mesoporous) microparticles or microspheres.

As demonstrated in our previous work [13], MP microparticles actually consist of microcapsules with a mesoporous silica shell, whose thickness and mass density ($\rho_{shell}$) were found to be about 1 μm and 0.83 g/cm^3^, respectively. Comparatively, if we assume a similar microcapsular structure for NMP microparticles, we would find an average mass density of about $\rho_{P123^{-}}=4.5$ g/cm^3^, which cannot be accepted, due to the total amount of silica encapsulated initially in each droplet (≃0.33 pg) and the known standard value of the mass density of silica, i.e., 2.2 g/cm^3^. Therefore, in order to match a reasonable value for the mass density of NMP microspheres, one should assume that each single P123^−^ droplet will split into at least two or three daughter droplets, each of which would lead to an NMP microparticle with half or a third of the total amount of silica encapsulated in the initial (mother) microdroplets. In such a way, the average maximum mass density of the resulting P123^−^ microparticles should be about 2.2 g/cm^3^ for a division by a factor of 2, or 1.5 g/cm^3^ for a division by a factor of 3. Furthermore, a division of the droplet volume by a factor of 4 may be envisaged, for which one may obtain a mass density for NMP microparticles of about 1.1 g/cm^3^. However, such divisions are not favourable from a surface tension point of view, as the total interfacial energy of three daughter droplets, for instance, is basically larger than the interfacial energy of their mother droplet, by a factor of $3^{\left(1/3\right)}=1.44$. Nevertheless, our optical microscopy observations, performed during the condensation of P123^−^ droplets, shows that such divisions indeed occur. We observe in Figure 3 the extrusion of small droplets from initial ones during the condensation process. We also observe the formation of bridges between these different droplets, which allows the droplet contents to homogenise and to average their size during condensation. Interestingly, Figure 4 also shows three SEM images where we can see two, three and four connected NMP microcapsules, as suggested above. These microparticles were obtained from single isolated mother droplets. We suggest that the droplet-stabilising surfactant molecules used may not distribute homogeneously around the droplet, caused for instance by the tip-streaming effect [22,23]. This effect may lead to instabilities of the droplet interface, through which the droplet content may be expelled and form smaller daughter droplets.

In order to better understand the effect of the P123 meso-structuring agent on the formation of silica microparticles and their density, we performed small-angle X-ray scattering (SAXS) and BET (Brunauer, Emmett, Teller) measurements [24], as shown in Figure 5 and Figure 6, respectively.

The SAXS diagram of MP microspheres, shown in Figure 5 (solid black curve), exhibits three significant peaks at 2θ = 1.2°, 1.8° and 2.1°. The observation of such peaks demonstrates clearly that mesopores of MP microspheres form a well-ordered hexagonal structure with the following lattice reticular distances: d[10] = 8.5 nm, d[10] = 4.9 nm and d[20] = 4.25 nm. In contrast, the SAXS diagram of NMP microparticles (red dashed curve in Figure 5), shows no diffraction peaks. This shows clearly that pores obtained with this sample are not ordered.

BET isotherm diagrams of both mesoporous and non-mesoporous samples, shown in Figure 6, give more detailed information about the mass density of these samples and their type of nanopores. According to IUPAC, the observed isotherm diagrams are of IV type for the MP sample (red curve in Figure 6) and of V type for the NMP sample (blue curve in Figure 6). They indicate the presence of cylindrical mesopores with bimodal pore openings, i.e., pores are open at both ends for mesoporous microspheres. The pore-size distribution calculated from the adsorption isotherm is comparatively narrow and has a distribution maximum at a pore diameter of 5.9 nm, a specific surface of *S_BET_* = 514 m^2^/g and a pore volume of $V_{P}=0.76$ cm^3^/g (see Table 1). Comparatively, the analysis of the recorded hysteresis isotherm of NMP microspheres (blue curve) reveals that this sample has a low porosity with an average pore diameter of about 13.5 nm, a relatively small specific surface, $S_{BET}=80$ m^2^/g and a pore volume $V_{P}=0.27$ cm^3^/g, as shown in Table 1. Following a simple calculation, detailed below, the measured value of the specific volume of pores ($V_{P}$) also enables an experimental value for the mass density $\rho_{NMP}$ of NMP microspheres to be deduced, which is found to be about 1.4 g/cm^3^. Indeed, since 1 g of silica occupies a volume $V_{SiO_{2}}=1/\rho_{SiO_{2}}=0.45$ cm^3^/g and the specific pores volume is $V_{P}=0.27$ cm^3^/g, the overall volume occupied by both silica and the empty pores (per mass unit) should be equal to $V_{SiO_{2}}$ + $V_{P}$ = 0.72 cm^3^/g. We therefore deduce an experimental value for the mass density of non-mesoporous silica microspheres of about $\rho_{P123^{-}}=\frac{1}{0.72}=1.4$ g/cm^3^. This value is very close to the third of the previously calculated mass density value (4.5 g/cm^3^) for an NMP microsphere that would have been obtained if these droplets did not split into smaller droplets.

### 3.2. Suggested Mechanisms for the Formation of Mesoporous and Non-Mesoporous Silica Microspheres

In order to unravel the formation mechanism of microfluidic-based silica microspheres, we conducted measurements of the size of P123^−^ and P123^+^ microdroplets versus time during their condensation process, as shown in Figure 7. We notice that the kinetics of the condensation of P123^−^ microdroplets is very different from that previously observed with P123^+^ microdroplets [13] (see inset of Figure 7 and [13] for a detailed description). Indeed, in the presence of the P123 agent, the size of P123^+^ microdroplets remains practically constant during a relatively long period of time (approximately one hour), then droplet size starts to decrease linearly with a constant rate of about $K_{+}≃-0.13$ μm/min, before MP microspheres reach their final size (10 μm) after an overall time period of ∼2.5 h. The observed linear decrease in the size of P123^+^ microdroplets may be explained easily using a simple model based on a solvent evaporation process, as illustrated in Figure 8A. Indeed, the change versus time of the volume *V* of a droplet with a radius R(t), at a given time *t*, may be reasonably assumed to be proportional to the area of the droplet surface in contact with air (since the solvent does not dissolve in the surrounding fluorocarbon oil). Therefore, one can write
(1)$$\frac{dV}{dt}=4\pi R^{2}\frac{dR}{dt}\propto -R^{2}$$

Equation (Equation 1) leads to
(2)$$R\left(t\right)=K_{+}t+R_{0}$$
where $K_{+}$ represents the slope of the linear decrease in the size of the droplets due to the solvent evaporation process, and $R_{0}$ represents their average initial size.

In the absence of P123 surfactant molecules, we observe that the size of microdroplets starts to decrease immediately after droplets reach the Petri dish where they are collected, as shown in Figure 7. We notice also that the change of the size of P123^−^ microdroplets versus time cannot be fitted in this case with a linear function as for P123^+^ microdroplets. Before a detailed discussion about the change of the size of P123^−^ microdroplets versus time, let us first discuss the role played by P123 surfactant molecules in the division process of microdroplets (when they are present). We suggest indeed that, driven by the solvent evaporation, the division of P123^−^ microdroplets starts soon after the droplet’s radius reaches a critical value, $R_{c}$, below which internal Laplace pressure, $\Delta P=\frac{2\gamma }{R}$, becomes larger than the droplet interfacial tension. This leads to the extrusion of a part of the droplet’s content and eventually gives birth to new droplets and ultimately to smaller silica microspheres.

In contrast, when P123 surfactant molecules are present, they may organise at the droplet interface, and enable silica aggregates to self-organise and to solidify around cylindrical channels built by P123 meso-structuring molecules at the droplet interface (see Figure 8 and Figure 9). Therefore, when droplets reach the critical size $R_{c}$, a rigid mesoporous silica shell has enough time to build and to solidify, thus enabling it to withstand the increasing Laplace pressure following solvent evaporation and lowering of the droplet’s radius.

The observed profile of P123^−^ microdroplets size versus time, shown in Figure 7, may be explained by considering both solvent evaporation and division of droplets. Furthermore, as a consequence of a natural (Gaussian) distribution of the density of the pores and their type, i.e., open at both ends or closed at one of their ends, the rate of the evaporation process across the forming pores and shell should also obey a similar (Gaussian) distribution, and also the time at which droplets reach their critical radius, $R_{c}$, should be $t_{c}$, as illustrated in light brown in Figure 10.

Since the division is occurring on the distribution scale, the change of droplet diameter could be represented by an error function (erf), which corresponds to the integral of the Gaussian distribution, drawn in purple and brown lines in Figure 10A, where we present two situations. An idealised one with a sudden division at $R_{c}$ and a more realistic one with a time duration representing the non-homogeneity of the local evaporation rate and critical radius $R_{c}$ (or a critical diameter $D_{c}$).

The two represented evaporation models (blue and red lines) correspond to the two chosen distributions: (i) the idealised one (σ=1) with the thin Gaussian distribution (brown) and corresponding to a step division (erf function) and (ii) the realistic case (σ=15) with a larger Gaussian distribution and a larger division duration time (purple line). Such idealised evolution is represented with a grey colour zone in Figure 10B for a unique evaporation rate. It corresponds to the blue thin Gaussian distribution curve (σ=1) and exhibits a distinguished linear evaporation followed by a sudden division when the radius reaches the critical value ($R_{c}$ ), then a second linear evolution starts due to a second step of solvent evaporation, as illustrated in Figure 10B (blue magenta dot line). Hence, one should observe, as for the ideal case, a first linear decrease in the size of the droplet with a slope $K_{evap}=-\alpha$ until the radius R(t) reaches a critical value $R_{c}$ (or a critical diameter $D_{c}$, as represented with the blue curve in Figure 10B), followed by a second linear decrease in the size the droplet with a smaller slope.

We may determine the value of the first evaporation rate by considering the tangent line to the experimental curve at the origin. We find α≃0.3 μm/min, which enables in turn the corresponding critical diameter and critical time values (for a monodisperse evaporation rate) to be determined. We find $D_{c}≃23$ μm and $t_{c}≃30$ min, respectively. It is worth noting that α value is approximately two times larger than the experimental values observed for the solvent evaporation rate of P123^+^ droplets ($|K_{+}|≃0.13$) and the second evaporation rate, labelled β in Figure 10. This result is in a good agreement with our model formation of an organised layer of P123 molecules at the droplet interface (illustrated in Figure 9), which contributes to the slowing down of the solvent evaporation rate and gives enough time for the formation of a stable silica shell around the droplet interface that can withstand the increasing droplet internal Laplace pressure. This statement is also supported by the results of a recent study carried out by Wang et al. [25], about the effects of the SDBS (sodium dodecyl benzene sulfonate) surfactant and HPAM (hydrolyzed polyacrylamide) polymer on the formation and stabilisation of oil-based foam liquid films, where the authors report that the stability of their emulsions was enhanced upon the increase in SDBS and HPAM concentrations.

The experimental evolution versus time of the average droplet’s diameter and its modelisation with the more realistic model, which takes into accounts a larger Gaussian distributions of evaporation rate and critical time $t_{c}$ droplet division, is represented with the red line curve in Figure 10. It is interesting to note that between the two evaporation regimes (1 and 2 in Figure 10B), the average radius of a droplet, R(t), should undergo a sudden jump until a value $∼\frac{R_{c}}{\sqrt[{3}]{3}}≃0.7R_{c}$ is reached. The theoretical value of such a jump (0.70) is found to be very close to the one determined (0.65) from Figure 10B. These results confirm, as suggested earlier, that in the absence of P123 molecules, droplets divide on average into three droplets before giving birth to smaller NMP silica microspheres with an average mass density of about 1.4 g/cm^3^. The division of the droplet’s volume by a factor of 3 should be considered as an average situation between a division by a factor of 2 and a factor of 4, as illustrated in Figure 4.

## 4. Conclusions

We have shown in this study that, besides offering a unique tool for the fabrication of well-defined monodisperse mesoporous silica hollow microspheres (microcapsules), whose size, shape and composition can be varied on demand, droplet microfluidics enables a better understanding of the role and effect of different parameters, such as the addition of Pluronic P123^®^ meso-structuring surfactant, which is generally used for the control of the porosity of mesoporous silica materials. We have shown in particular, that although both types of initial precursor droplets, prepared with and without P123 meso-structuring agent (namely P123^+^ and P123^−^), have a similar diameter (≃30 μm) and TEOS silica precursor concentration (0.34 M), the resulting microparticles exhibit two noticeably different sizes: 10 μm and 5.2 μm for P123^+^ and P123^−^ microparticles, respectively. Structural features of P123^−^ microparticles obtained in the absence of Pluronic molecules are explained by the microdroplets occurring during condensation, below a critical size, from a division process of the initial microdroplets into three smaller daughter microdroplets before they fully condense into solid silica P123^−^ microspheres. Based on the different results we obtained from SEM, SAXS and BET, we developed an original model for the formation mechanism of silica microspheres in the presence and in the absence of P123 molecules. This model is based on the combination of both Gaussian distributions of the solvent evaporation and the division of droplets, which explain the shape of the change of droplet size versus time during the condensation process. We suggest that for P123^+^ microdroplets and resulting mesoporous microparticles, P123 surfactant molecules organise at the droplet interface, enable silica aggregates to self-organise and to solidify around cylindrical channels built by the meso-structuring molecules at the droplet interface. In such a manner, when droplets reach a critical size Rc, below which the internal Laplace pressure becomes larger than the droplet interfacial tension, a rigid mesoporous silica shell has enough time to build and to solidify, enabling it to withstand the increasing Laplace pressure, which is driven by the solvent evaporation and the lowering of the droplet radius of the condensing microdroplets.

## Figures and Tables

**Figure 1 micromachines-14-00936-f001:**
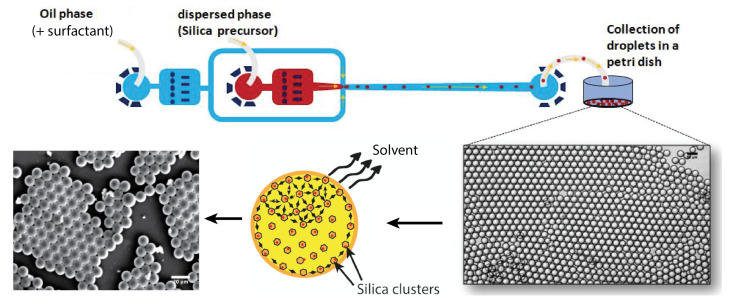
Design of the microfluidic device and droplet collection.

**Figure 2 micromachines-14-00936-f002:**
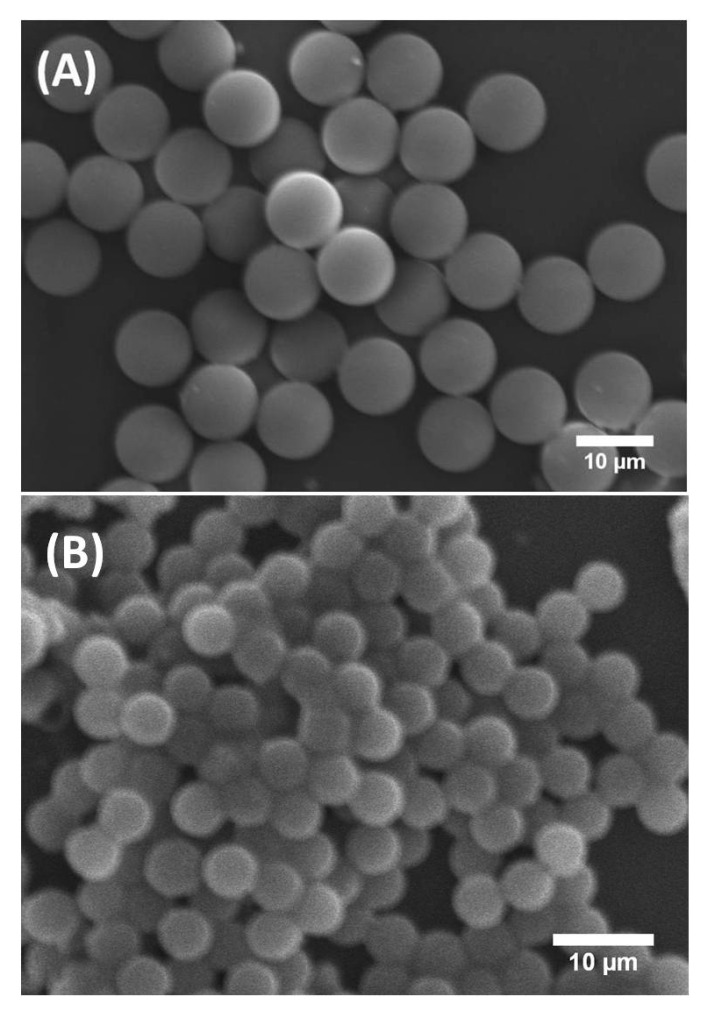
SEM images of mesoporous (MP) silica microspheres (**A**) and non-mesoporous (NMP) silica microspheres (**B**).

**Figure 3 micromachines-14-00936-f003:**
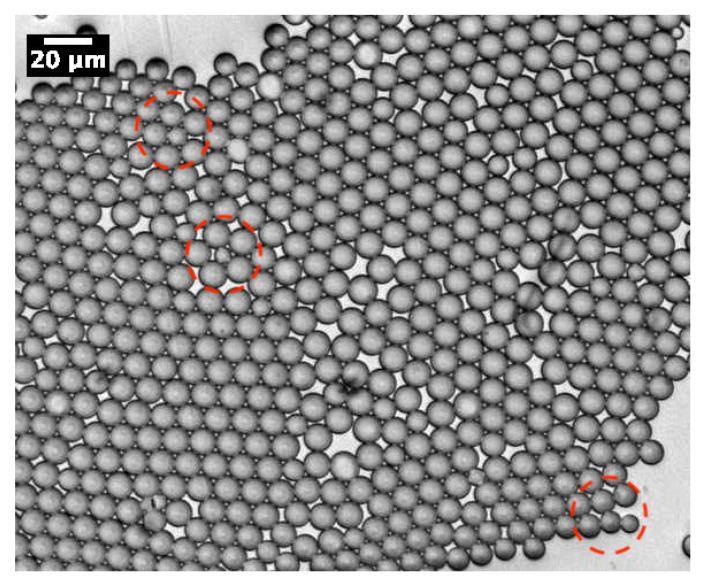
Optical micrograph of P123^−^ droplets during their condensation, which shows the formation of daughter droplets by extrusion of the content from initial droplets (red dashed zones). We observe also the formation of bridges between microdroplets enabling homogenisation of the content and size of droplets during their condensation.

**Figure 4 micromachines-14-00936-f004:**
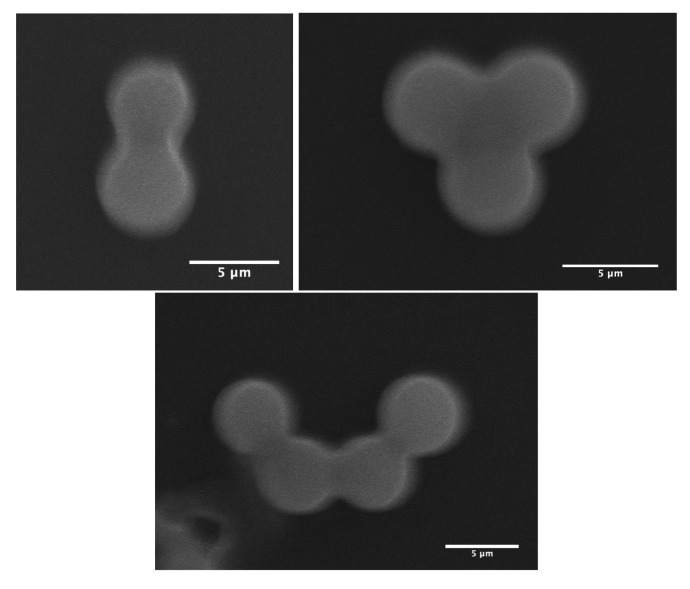
SEM images of two, three and four NMP microparticles derived from isolated droplets. Scale bars = 5 μm.

**Figure 5 micromachines-14-00936-f005:**
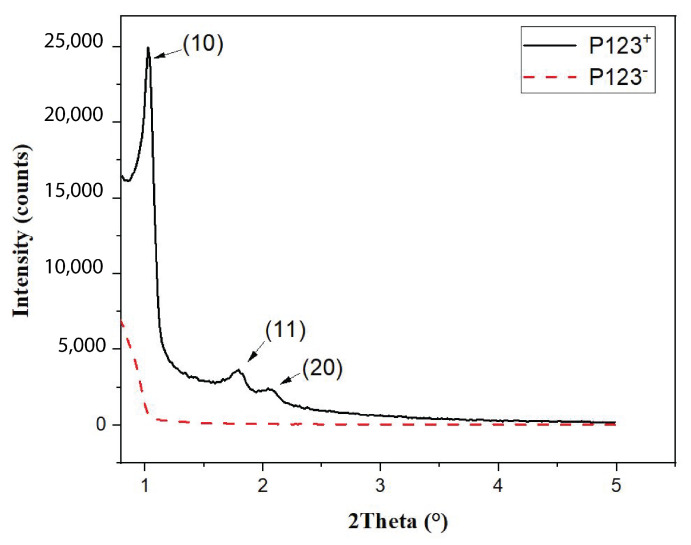
Small-angle X-ray diffraction pattern showing the ordering of hexagonal array of nanopores in mesoporous silica (P123^+^) microspheres and its absence in the non-mesoporous ones (P123^−^).

**Figure 6 micromachines-14-00936-f006:**
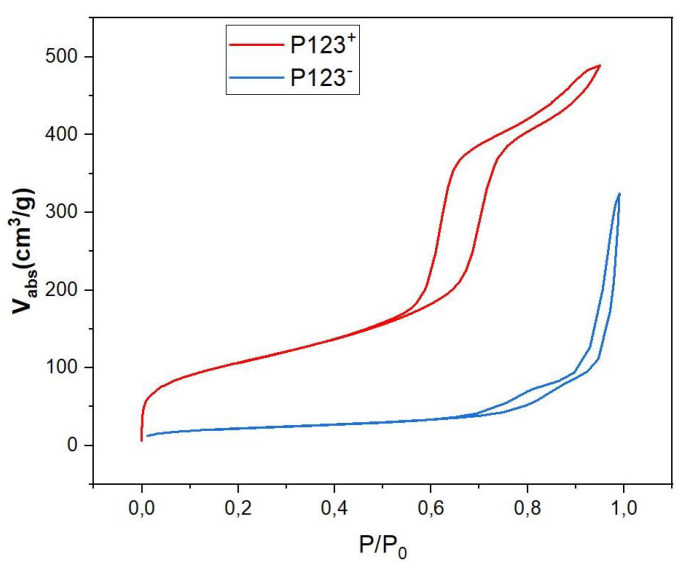
*N*_2_ adsorption/desorption isotherms of P123^+^ (red curve) and P123^−^ (blue curve) samples.

**Figure 7 micromachines-14-00936-f007:**
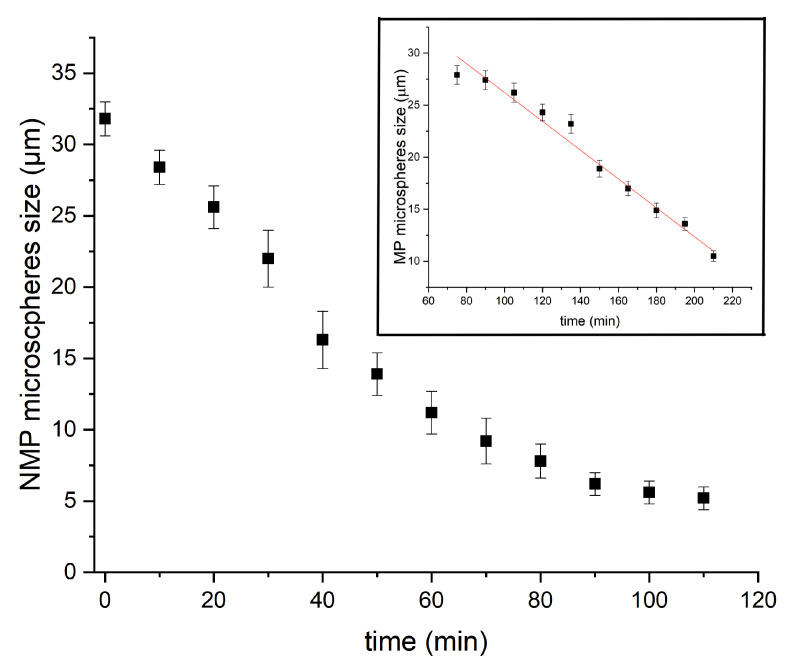
Kinetics evolution, from liquid drops to solid microparticles, with/without P123.

**Figure 8 micromachines-14-00936-f008:**
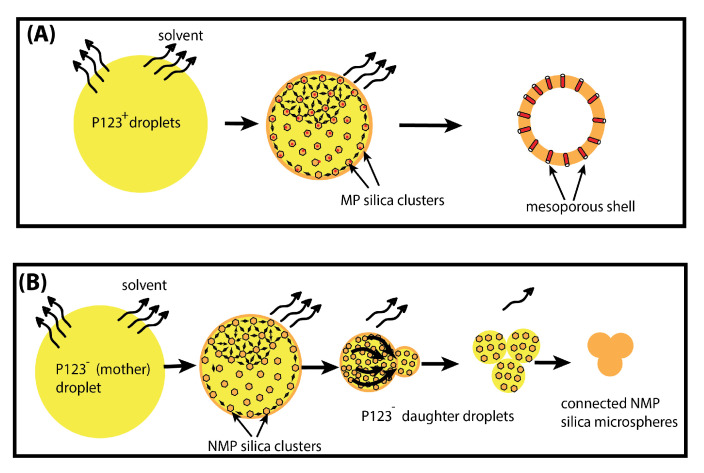
Schematic illustration of the formation of mesoporous (**A**) and non-mesoporous (**B**) microparticles, obtained with and without P123 meso-structuring agent, respectively.

**Figure 9 micromachines-14-00936-f009:**
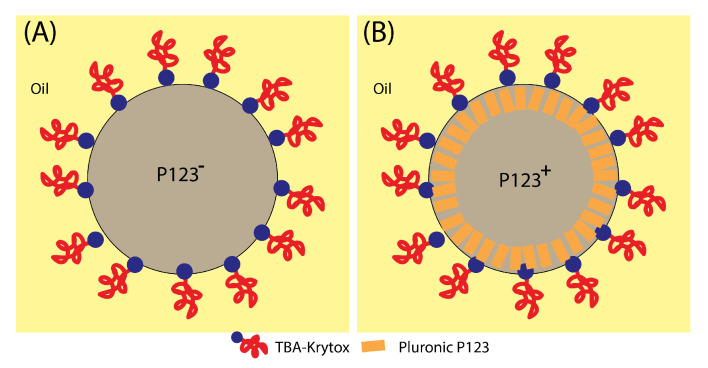
Schematic illustration of the organisation of silica meso-structuring surfactant (Pluronic P123) and droplet-stabilising surfactant (TBA-krytox) interfacial organisation. Addition of Pluronic P123 (**A**) increases the droplet interface rigidity and enhances its stability against droplet internal pressure and a subsequent division and the blobbing of a mother droplet, which leads in average to 3 daughter droplets when Pluronic P123 surfactant is missing (**B**).

**Figure 10 micromachines-14-00936-f010:**
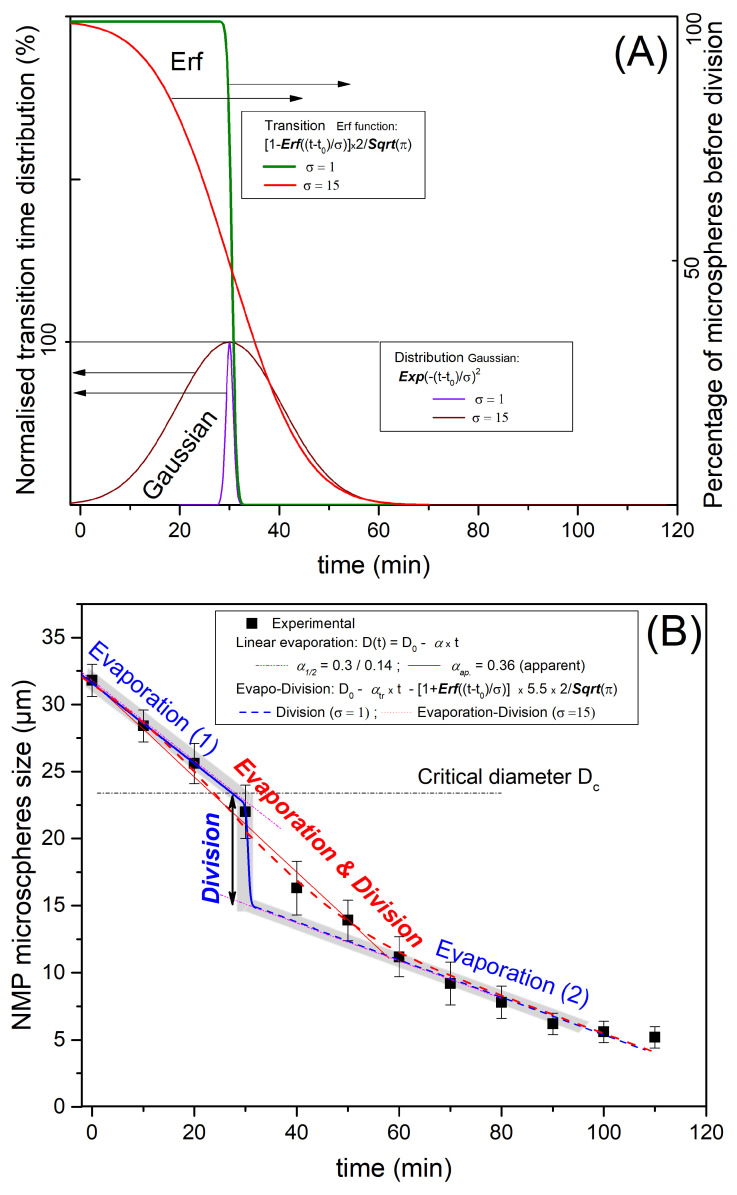
P123^+^ and P123^−^ droplets condensation kinetics models combining evaporation and division. In Figure (**A**), we consider both an idealised narrow Gaussian division distribution (σ=1, purple curve) and a more realistic larger Gaussian distribution (σ=15, brown curve), which may be attributed to fluctuation of the diameter and irregularities in local evaporation and also the local interface instabilities. The corresponding transition curves of the droplets’ average size versus time are deduced from error functions plots (green curve for σ=1 and red curve for σ=15). In Figure (**B**), we compare the obtained experimental results (droplets size versus time) with model results for the idealised abrupt division (blue line) separating the two (early and late) evaporation processes occurring at a critical microsphere diameter (*D_c_*). We plot the second, more realistic evolution (red curve) for which droplet division occurs on a larger time scale (∼30 min). For the narrow distribution model, i.e., short time duration of division, evaporation and division may be considered separately, whereas for large division distribution, evaporation and division should be considered simultaneously. This explains the combined intermediate evolution (evaporation and division) plotted with the red dotted line.

**Table 1 micromachines-14-00936-t001:** Textural parameters of mesporous (MP) and non-mesoporous (NMP) microspheres obtained from 32 μm droplets; S_*BET*_, *V*_*P*_ and *D*_*P*_ represent specific surface area, mesopore volume and mesopore size of MP and NMP samples, respectively.

	$S_{\mathrm{BET}}$ (m^2^/g)	$V_{P}$ (cm^3^/g)	$D_{P}$ (nm)
MP	514	0.76	5.9
NMP	80	0.27	13.5

## Data Availability

Not applicable.

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
