# Peer review of "Role and Effect of Meso-Structuring Surfactants on Properties and Formation Mechanism of Microfluidic-Enabled Mesoporous Silica Microspheres"

_micromachines, 2023, doi:10.3390/mi14050936_

Round 1

Reviewer 1 Report

Bchellaoui and co-workers studied the impact of adding P123 meso-structuring surfactant to increase the droplets' interface rigidity. They hypothesized that the P123 will form cylindrical pores, generating a mesoporous shell around the droplet. To compare the possible impact of adding P123, they investigated the droplets without adding P123 and then build a model to measure the evaporation of the droplet. In this study, they used a microfluidic setup to generate droplets. The overall content of the paper is appealing. However, after careful consideration, I would like to share the following comments/suggestions with the authors for further consideration.

1) Introduction: "A crucial step of our approach relies on the control of the solvent evaporation process which is carried outside the microfluidic channels." That would be worth explaining a little bit about the previous method here.

2) section 2.1.: "Diffusion and dispersion of both solvents and sol can may be neglected during the flow of droplets along the microfluidic channel and the tubing. " In the absence of any advection, why we can ignore diffusion and dispersion?

3) Same section: "Consequently, the condensation of silica is mainly governed in our study by a slow evaporation process, at room temperature, of the solvents at the oil-air interface." How can we infer this conclusion from the previous sentence that explained the ignoring of the diffusion of the solvent and sol?

4) Section 3.2. "In the absence of P123 surfactant molecules, we observe that the size of microdroplets starts to decrease immediately after droplets reach the Petri dish where they are collected, as shown in Fig. 7." The same thing does not happen for the P123+? In the image below, we can see that the solution of MP and NMP vaporization rates are almost the same at the initial times. The next question is why for the MP scenario, the time axis is started from 60 min. A final suggestion for this figure, it would be much more informative if both data sets are put together in a single plot.

5) In conclusion, it is stated that in this study a unique fabrication method is presented which can control the size, shape, and composition of the silica hollow microspheres. However, to my best understanding, this study does not suggest how we can change/control the size of the silica hollow microspheres. If the authors' intention is to justify this sentence by the different sizes of the P123+ and P123- particles, I think that is not the way to control the size of the particles, or at least authors could suggest how can we control the size of the particles with presenting particle sizes other than the 10um and 5um sizes.

6) Fig 10: It is really difficult to understand the message that this figure wants to convey. So, I strongly suggest that the whole figure be re-designed by adding a clearer legend and also a standalone caption. 

7) As a final question, it would be interesting for the readers to understand what is the physics behind adding P123 and having a mesoporous shell.

In my opinion,  the writing of this study is excellent and could convey the message very clearly. 

Reviewer 2 Report

The idea of the article is good and the experiment was well designed and performed. In spite of that, the work lack some improvements as follow:

1) Quantitative results must be included in abstract.

2) The introduction can be enriched. Authors should create a table that shows the novelty of this work.

3) section 3.2: I believe the following paper can be reviewed and support the mechanism explanation: “Effects of the Surfactant, Polymer, and Crude Oil Properties on the Formation and Stabilization of Oil-Based Foam Liquid Films: Insights from the Microscale(2022). J Molecular Liquids, 373, 121194”.

4) Please put enough emphasis on the points of novelty of the proposed study in Conclusions.
